# Developing a hope-focused intervention to prevent mental health problems and improve social outcomes for young women who are not in education, employment, or training (NEET): A qualitative co-design study in deprived coastal communities in South-East England

Clio Berry [1]*, Julia Fountain[2], Lindsay Forbes[3], Leanne Bogen-Johnston[4], Abigail Thomson[4], Yelena Zylko[4], Alice Tunks[1], Sarah Hotham[3], Daniel Michelson[4,5,6]

1 Department of Primary Care and Public Health, Brighton and Sussex Medical School, University of Brighton, Falmer, Brighton, United Kingdom, 2 Research and Development, Sussex Partnership NHS Foundation Trust, Sussex Education Centre, Hove, United Kingdom, 3 Centre for Health Services Studies, University of Kent, Canterbury, Kent, United Kingdom, 4 School of Psychology, University of Sussex, Falmer, Brighton, United Kingdom, 5 Department of Child & Adolescent Psychiatry, Institute of Psychiatry, Psychology and Neuroscience, King's College London, London, United Kingdom, 6 NIHR Maudsley Biomedical Research Centre, South London and Maudsley NHS Foundation Trust and King's College London, London, United Kingdom

* c.berry@bsms.ac.uk

## Abstract

Young women who are not in education, employment, or training (NEET) experience poorer health and social outcomes compared to non-NEET young women and to NEET young men, especially in deprived areas with intersecting inequalities. The evidence on effective public health approaches is scarce. Interventions that target hope, which NEET young women notably lack, offer a promising theory-driven and intuitive means to prevent mental health problems and improve social outcomes. Hope can be defined as a goal-focused mindset comprising self-agency (motivation and self-belief) and pathways (identifying routes to achieving goals). Hope is implicated in a variety of evidence-based psychosocial interventions for young people, but is not directly targeted by existing prevention programmes for NEET populations. The current study used a phased qualitative research design and participatory methods to model a hope-focused intervention for NEET young women. Phase 1 investigated population needs and intervention parameters through semi-structured interviews with 28 key informants living or working in disadvantaged coastal communities in South-East England. The sample comprised eight NEET young women, four family members, and 16 practitioners from relevant support organisations. Phase 2 refined intervention parameters and outcomes through co-design sessions with four NEET young women, followed by a theory of change workshop with 10 practitioners. The resulting intervention model is articulated as a mentor-supported, in-person psychosocial intervention that builds

**Data Availability Statement:** Data in the form of collected interview transcripts are not shared publicly due to ethical concerns regarding potential identifiability. These interview transcripts were collected from a small number of individuals, from specific populations within identified geographical regions. Participants were not asked to consent specifically to the public sharing of their verbatim transcripts in full. However, the full analysis with raw data in the form of verbatim quotes is presented as Supporting information and this analysis presents all relevant data. Full data for Phase 2 co-design participant rankings of proposed intervention components are presented Supporting information.

**Funding:** This study was funded by an Application Development Award (reference NIHR135316) from the Public Health Research Programme of the UK National Institute for Health and Care Research (NIHR; https://www.nihr.ac.uk/explore-nihr/ funding-programmes/public-health-research.htm), awarded to DM, CB, JF and LF. The views expressed are those of the authors and not necessarily those of the NIHR or the Department of Health and Social Care. The funders had no role in study design, data collection and analysis, decision to publish, or preparation of the manuscript.

**Competing interests:** The authors have declared that no competing interests exist.

hope by enhancing positive sense of self and time spent in meaningful activities, before explicitly teaching the skills needed to identify, set, and pursue personally meaningful goals.

## Introduction

Young people who are Not in Education, Employment, or Training (NEET) are a cause of major public health concern. In the UK, they account for approximately 14% of 16–24-year-olds [1]; globally this figure rises to approximately 1 in 5 young people, with young women significantly more likely to be NEET [2]. NEET young people experience significantly worse health outcomes and life chances than economically active and socially engaged peers [3]. Mental health problems are especially prevalent, with anxiety and depression around twice as common in the NEET group compared with young people generally [4]. Being NEET additionally puts young people at elevated risk of sequelae such as self-harm and psychiatric hospitalisation [4–6], and robustly predicts transition from mild mental health symptoms to disorders of diagnostic severity [7, 8]. 'Scarring' effects are additionally observed, such that time spent NEET in adolescence is associated with chronic mental health problems, unemployment, and low wages over 20 years later [9, 10].

In the UK, as globally, the number of NEET young women is increasing [1]. Risk factors and outcomes for NEET status show marked gendered inequalities. Young women have worse mental health than male peers at school, and poorer adjustment over post-school transitions [11]. Young women have a higher risk than young men of becoming NEET in the context of family unemployment, deprived backgrounds, when they themselves have poor educational achievement, and when they have caring responsibilities [12–17]. Once NEET, UK data [12] demonstrates that, compared to non-NEET young people and to NEET young men, NEET young women face complex constellations of cumulative disadvantage, including greater likelihood of chronic health conditions, comorbid health problems. NEET young women, compared to NEET young men, are more likely to experience psychological distress [18, 19] poorer mental health [20], greater suicidality and self-harm [21]. Socio-occupational outcomes are poorer too in this group. NEET young women, relative to young men, experience reduced life satisfaction and sense of control [18], a reduced sense of positive identity [18], experience greater social isolation [13], and poorer social and relationship outcomes [18], with greater risk of sexual violence [12]. NEET young women are also more likely to encounter discouragement and stigma from potential employers, with a lack of flexible occupational opportunities that fit around caring responsibilities [13, 22]. These multiple disadvantages are compounded for NEET young women in deprived areas [23], including in many coastal regions around the UK, which have relatively fewer jobs, under-performing schools, and poor infrastructure relative to in-land areas [24]. Poorer outcomes for young women, in turn, increase the risk that NEET status will persist over time [25].

Research and interventions specifically focused on NEET young women, especially in the UK context, remain relatively scarce, despite significant policy interest on the social welfare implications of being NEET [26]. One study reported on a qualitative evaluation of a computer coding intervention for NEET young women in England [27], showing promise with respect to enhancing sense of positive identity but not demonstrating wider health or social benefits. The research gap for NEET young women is especially problematic given that there is an ongoing crisis of youth support provision in the UK, especially in deprived areas such as the coast [28]. Even if youth mental health or social support services are available, NEET

young people are less likely to be able to access and engage with them [29]. Moreover, generic psychosocial interventions appear less engaging and effective for the most disadvantaged NEET subgroups [17].

Whilst evidence suggests that NEET young women are more likely to experience prior and ongoing adversities, structural factors do not wholly dictate young people's occupational trajectories [30]. Recent studies suggest that the greater risk of being NEET, and of having poor outcomes for different demographic groups, is at least partially explained by attitudinal differences [31, 32]. There is also evidence for a reciprocal relationship between NEET status and mental health symptoms, where the perception (from oneself and others) of failing to 'keep up' can be harmful to mental health and discourage service use and social re-engagement [31, 32]. Subjective perceptions related to self-worth and capacity to reach personal goals, therefore, represent important intermediate outcomes, especially among young people for whom immediate re-entry into education, training or work may be unrealistic [30].

Such self-perceptions can be understood as manifestations of the theoretical construct of hope. Hope can be defined as a way of thinking, which comprises self-agency (the 'will' or motivation and self-belief) and pathways (the 'ways' or goal route identification) directed towards a meaningful goal [33]. Hopeful self-agency predicts positive youth education and work outcomes [34, 35], predicting success even more strongly than intelligence and cognitive ability [36], and protects against the effects of structural and social disadvantage and adversity [30, 37, 38]. NEET groups commonly lack self-belief and optimism about changing their circumstances, even when they are motivated to seek work or education [4]. Diminished hope is compounded by the projection of low expectations from family members and others [39], reflecting the intergenerational and systemic transmission of low hope [30]. A recent UK study found that NEET young people exhibited less hopeful self-agency compared with other groups, while greater self-agency reduced the longitudinal risk of being NEET [30]. NEET young women are again doubly disadvantaged: viewing themselves as less competent than NEET young men and reporting lower professional expectations [40]. Gendered differences are exacerbated in smaller, rural, and remote communities, in which young women experience a reduction in hope during late adolescence whilst young men experience growth [41].

Hope has been strongly implicated as a common mechanism in a variety of talking therapies [42], yet has been overlooked with respect to interventions for NEET young people. A previous systematic review concluded that there is a substantial gap in knowledge about what aspects of re-engagement interventions work or how [17]. This research agenda has been complicated by the confounding effects of multi-agency working, such that several organisations commonly interact with NEET young people at any one time [17]. There is a particular dearth of research on theory-driven interventions for the NEET population [17]. A focus on hope offers an opportunity to create a theory-driven approach to augment gendered inequalities that render young women more likely to become and stay NEET, with associated poor health and low aspirations for the future.

In a recent systematic review [42], we found that effective hope-focused interventions for youth depression commonly incorporated cognitive, behavioural, and interpersonal elements that (i) promoted hopeful thinking; (ii) identified meaningful goals and activities; and (iii) were delivered in a timely, hopeful, and inspiring way. The constituent practice elements were often brief (i.e., involving six or fewer sessions) and included (but were not limited to) structured goal setting, teaching problem-solving skills, and supporting young people to visualise positive futures. Our review additionally found that hope-enhancing interventions can be flexibly deployed, with evidence of effectiveness across health, social services, education, and community settings. Therefore, a hope-focused intervention approach has the potential to offer an efficient and scalable solution to promoting positive outcomes for NEET young women.

Moreover, a hope-focused stance is directly relevant to managing service engagement challenges associated with NEET groups, for hopefulness is implicated in increased help-seeking and intervention engagement in youth [42]. Engagement may be strengthened further by using other best practices identified from NEET-focused case studies [43]. These include: (i) offering flexible support with a high degree of personalisation; (ii) delivering programmes in 'neutral' community settings, without the stigma or other negative connotations of statutory services; (iii) encouraging non-hierarchical relationships between participants and providers; (iv) providing individual support for young people who experience multiple barriers to re-engagement and/or have been NEET long-term; and (v) involving community representatives and NEET young people in the design and possible delivery of programmes.

From these conceptual and practice-based starting points, we aimed to develop a low-cost, scalable hope-focused psychosocial intervention for NEET young women aged 16–24 years living in areas of coastal deprivation. Our iterative developmental approach involved working alongside NEET young women and other stakeholders to identify local population needs and intervention requirements; select and contextualise promising hope-focused practice components; and articulate how a hope-focused intervention might be implemented and scaled up to impact positively on mental health and social functioning. The specific research questions were:

1. What formats and settings are viable for delivering a brief, low-cost, hope-focused intervention to NEET young women aged 16–24 years in deprived coastal communities?

2. Who are possible non-specialists that could credibly deliver the intervention?

3. What are the most meaningful short- and long-term outcomes related to hope and its theorised effects on mental health and social functioning, and how can these be measured?

4. What is the theory of change for the intervention?

## Materials and methods

### Research design

We applied a phased, qualitative design that drew on principles and practices of person-based intervention development [44] and partnership [45], i.e., incorporating the perspectives of the people who will use the intervention and the intended beneficiaries within design decisions. We took a critical realist stance [46, 47], reflecting the premise that an objective reality exists (a realist ontology) but is imperfectly accessed through a lens of individual perception (a relativist epistemology). An iterative process of reflection and action entailed capturing local needs and priorities in formative interviews with varied stakeholders from coastal areas (Phase 1), followed by co-design sessions with NEET young women (Phase 2). We conceptualised co-design as involving intended end beneficiaries directly in defining key intervention parameters, with an emphasis on reciprocal democratic dialogue and equal participation [48]. In Phase 3, we conducted a Theory of Change (ToC) workshop with local practitioners. Finally, findings from primary data collection were synthesised along with pre-existing evidence from a recent systematic review [42], and additional scoping reviews, to inform the creation of an intervention 'blueprint' based around standardised intervention descriptors [49]. This approach is appropriate for critical realist research, i.e., considering data through the original premise of the study and adding further theories as indicated by iterative data gathering and analysis [46]. The study was overseen by an independent steering committee, representing members with scientific, clinical and/or lived expertise pertaining to youth and women's mental health. This

report has been developed in line with the GUIDED approach to reporting health intervention development [50].

## Participants and setting

The study was conducted in two geographical regions in South-East England, Sussex and Kent, which contain socially and economically deprived coastal areas and higher than average populations of NEET young people at 16% each [1]. The study protocol was prospectively registered (17/06/2022; Research Registry reference 8016). Research ethics approval was obtained from the Health Research Authority (reference 22/HRA/0721, IRAS 310439) and the University of Sussex C-REC Committee (reference ER/DMM55/1). Participants provided informed consent using an online or paper-based consent form before taking part.

Phase 1 participants (N = 28) were sampled from three stakeholder groups: (i) 16-26-year-old women who self-identified as having lived experience of being NEET (n = 8) in the past 12 months (NEET status pertained to maximum age threshold of 25 years in some organisations in each study region); (ii) family members of this target group (n = 4); and (iii) practitioners from statutory service providers (e.g., public health, primary care, mental health services, social services, education and youth employment services) and community organisations (e.g., youth groups) (n = 16). Participants from groups (i) and (ii) were recruited by promoting the project on social media and advertising it in community spaces that the target group and/or their family members may visit (e.g., food banks and libraries). Group (iii) was recruited through email, telephone, and in-person contact attempts based on an initial scoping exercise of local services, followed by snowball sampling. Recruitment to Phases 2 and 3 involved inviting Phase 1 participants to continue their participation and seeking novel participants using the same strategies as in Phase 1. Youth participants in Phase 1 (n = 5, 62.5% Sussex and n = 3, 37.5% Kent) all self-identified as White and spoke English as a first language, and were primarily heterosexual, single, and living with parents. The mean (M) age was 20.13 years (standard deviation [SD] = 2.36). All young people described themselves as having experienced mental health problems and three (37.5%) identified an additional non-mental health-related disability (e.g., autism). Time spent NEET was variable: ≤6 months (n = 2, 25%); 7–12 months (n = 7, 50%); >12 months (n = 2, 25%). The four participating family members (n = 3, 75% Sussex; n = 1, 25% Kent) were mothers of NEET young women, two of whom lived with the young woman. These family members all self-identified as White and spoke English as their first language. The 16 practitioners involved in Phase 1 (n = 10, 62.5% Sussex; n = 6, 37.5% Kent; M age = 49.57 years, SD = 7.72; n = 9, 56.3% female) included psychiatric, primary care and public health practitioners, social, support and youth justice workers, and careers and employment advisors.

Phase 2 co-design sessions were conducted with n = 6 NEET young women, of whom three were involved in Phase 1. Due to an administrative error in correctly recording informed consent on two occasions, data were available from n = 4 NEET young women in Phase 2 (n = 3, 75% Sussex; n = 1, 25% Kent; M age = 18.75 years, SD = 2.87). Three of these four included participants had additionally participated in Phase 1. The Phase 3 ToC workshop included 10 new participants. Two additional practitioners (who had previously completed Phase 1 interviews) did not attend the workshop but contributed feedback by email. Two workshop participants also sent follow-up written feedback. Eight (66.67%) Phase 3 participants identified as female, the remainder as male.

## Procedures

**Phase 1.**   Semi-structured interviews (M duration = 51 minutes, range 31–69) were conducted mainly online (two involving NEET young women were in person) using topic guides

tailored to each stakeholder group. One interview with a NEET young woman was conducted with a peer researcher jointly involved alongside a non-peer researcher (see Table 1). More extensive involvement of peer researchers in interviews with young people was planned to improve rapport and richness of data but was ultimately not possible due to practical and personal circumstances affecting availability of trained peer researchers. Topic guides explored: (i) mental health and social problems experienced; (ii) links between contextual risk factors, hopefulness and trajectories of mental ill-health and social disability; (iii) desired outcomes for a hope-focused intervention; and (iv) facilitators and barriers to achieving these outcomes through current service provision. Additional questions for practitioners examined potential selection criteria and referral pathways for the putative intervention, as well as assets and existing practices that may support integration within local communities and service settings. All interviews were audio-recorded and transcribed verbatim.

**Table 1. Public community involvement and engagement (PCIE) roles and activities.**

| | Details | Phase 1 | Phase 2 | Phase 3 | Other |
|---|---|---|---|---|---|
| PCIE lead | • Expert in patient and public involvement in research, personal experience of being a parent to a young woman with experience of being NEET | • Involved in shaping phase 1 advertising and informational materials, and data collection materials and processes | • Involved in shaping phase 2 informational materials, and data collection materials and processes | | • Contribution to research design and embedded role in all project decision-making<br>• Used community contacts to help raise awareness of the study and stimulate recruitment<br>• Led the recruitment and appointment process for PCIE advisors and peer researchers, and supported and co-delivered their training<br>• Presented (with the first author) to the Youth Café PCIE consultation group<br>• Contribution to preparing manuscript for publication |
| PCIE steering committee members | • Two women with experience of providing personal and professional community support to NEET young women in deprived coastal communities. | | | | • Members of independently-chaired project steering committee |
| PCIE consultants | • Youth café—A youth-led group of young people, primarily female, who have experience in accessing mental health and other youth services, and of consulting on health research design and delivery [70]<br>• Parent of a NEET young person | •Youth café informed phase 1 advertising and informational materials, and data collection materials and processes | | | • The youth café and parent consulted reviewed a draft intervention blueprint. Reflections from the youth PCIE group and the parent PCIE consultant were integrated into the intervention blueprint (see integrated Results section for further details). |
| Peer researchers | • Young women with experience of mental health problems and accessing support services, and with experience of being NEET and/or providing support to young people | • Involvement in designing data collection activities<br>• One interview with a NEET young woman conducted with a peer researcher present<br>• Involvement in analysing interview data and identifying initial themes | • Involvement in designing data collection activities<br>• All co-creation sessions were facilitated by one or two peer researchers | • One peer researcher attended the workshop and contributed to reflective discussions | • Two peer researchers contributed to preparing manuscript for publication |

**Phase 2.** Co-design sessions were facilitated using Zoom videoconferencing by one or two peer researchers and lasted for M = 54 minutes (range 35–85). These sessions were conducted separately with each NEET young woman since all participants preferred to contribute individually rather than participating in the focus group format that we first offered. The involvement of peer researchers was intended to provide a degree of impartiality from the other research team members and encourage critical discussion of intervention concepts and plans. Building on insights from Phase 1, these sessions explored preferred intervention content, providers/supporters, modes of delivery, activities, timing, dosing, settings, and important subjective and objective outcomes. In each session, participants viewed a slideshow with an accompanying audio-recording that summarised the project aim and Phase 1 findings. Following this presentation, participants were invited to reflect on the presentation content and wider relevant literature (based on our earlier evidence synthesis of hope-focused interventions [42]). The session facilitators presented further slides that included a draft intervention blueprint and module outline (derived from an interim Phase 1 analysis involving peer and non-peer researchers), including potential delivery approaches and example activities. Co-design sessions were recorded and transcribed verbatim. After each session, participants were invited to take part in three polls administrated using the online survey software Qualtrics, in which they provided three sets of rankings related to the importance of potential intervention modules, activities within each module, and outcomes.

**Phase 3.** A 90-minute ToC workshop [51] was conducted online with practitioners using Zoom videoconferencing software. The workshop was used to consider and refine the intervention model, and delineate short, medium, and longer-term intervention outcomes and how these would be achieved. We additionally explored opportunities for how to deliver the intervention in the current youth support landscape. Participants in the ToC workshop (n = 10) were first presented with a slideshow providing a synthesis of findings from the earlier phases and then invited to provide feedback. The process of collecting feedback followed a 'backwards mapping' approach [51] where participants started at the end of the intervention to identify the outcomes, and then critically assessed how and why these outcomes would be achieved through the hope-focused intervention. These discussions were initially conducted in small groups, with feedback collated through an online whiteboard (EasyRetro), and subsequently explained and refined in the larger group. The workshop was recorded and transcribed verbatim. A draft ToC was shared by email with workshop attendees and two Phase 1 practitioner participants who were not able to attend. Email feedback was then incorporated into an updated model.

## Public and Community Involvement and Engagement (PCIE)

We embedded PCIE throughout this project to increase the relevance and quality of this research. Our reporting of PCIE activities is in line with the GRIPP2 framework [52]—identifying the methods, outcomes, benefits, and challenges of these activities. Our PCIE approach blended qualitative research with key informants together with PCIE and co-design activities [53]. Our research team included a PCIE lead (JF) with personal and professional experience of supporting NEET and vulnerable young women, who collaborated in the design and planning of the research study [54] and was a consistent presence in weekly research team meetings to embed a lived experience perspective in decision-making. In addition, PCIE advisors, peer researchers (YZ and AT), and PCIE consultants were involved in multi-dimensional project roles to ensure 'authentic collaboration' throughout the project lifecycle [55]. These roles and activities are described in Table 1. The outcomes of these activities included: shaping project informational materials to refer to hope as a 'changeable mindset' and use simpler language;

streamlining the focus of Phase 1 interview topic guides; ensuring that the data collection processes were maximally inclusive and flexible (Phases 1 and 2); informing the selection of the specific foci and materials for Phase 2 co-design sessions (i.e., mentor delivery model, intervention modules and content, primary and secondary outcome selection); contributing to Phase 1 and leading Phase 2 data collection; and contributing as equal (Phase 1), and primary (Phase 2) data analysis partners. PCIE activity findings are additionally detailed and collated within integrated findings (see Table 3).

## Data analysis

We used a deductive-inductive approach to data analysis, which enabled us to select from and integrate candidate intervention elements and delivery strategies (57,58). In keeping with our critical realist stance [46, 47], we were interested in trying to elucidate 'demi-regularities' [46, 47], i.e., points of convergence and divergence across stakeholder groups and project phases, and in considering the causal phenomena and wider social structures that may give rise to these patterns in meaning. These foci are important in the context of critical realism's relativist epistemology (*versus* its realist ontology) because multiple explanations for observed phenomena are possible [56].

Data were analysed iteratively for each phase, using a framework charting approach [57], and then integrated in a mixed methods matrix [58]. A framework analysis approach is appropriate for critical realist research because both prioritise meaning-making in the context of complex phenomena and position research (and its products) as purposively derived from the marriage of participant experiences and researcher interpretations [56]. Moreover, a framework analytic approach facilitates use of all data [57], including that which that does not "fit" with dominant narratives, but which nonetheless may be important [46, 56]. The analysis was driven by our research questions and structured around items from the TIDieR framework [49], in keeping with our aim to develop an intervention blueprint detailing key intervention parameters. First, we coded the data deductively using the research questions and then structured the codes using the TIDieR items. We then identified meaningful subthemes under each TIDieR item, or research question in absence of a relevant TIDieR item. Theory is central to critical realist research [56]. We situated *a priori* the intervention within the cognitive model of hope [33], in which hope represents future-oriented goal-directed cognition comprising self-agency and pathways thinking. For procedural components (within the TIDieR "what" category), we structured codes pertaining to a published taxonomy [59] of specific therapeutic activities and techniques: categorising them as behavioural, cognitive, interpersonal, and non-specific, adding an 'additional activities' category for miscellaneous items. Subthemes in other TIDieR categories (or pertaining to research questions otherwise) were inductively coded.

For Phase 1, an initial meeting with the entire research team, including two peer researchers (YZ and AT), was held to discuss preliminary themes and impressions of the interview transcripts. We further refined the analysis in subsequent meetings with a smaller group and the first author (CB) drafted the final analytic framework. For Phase 2, three team members (YZ, AT, and CB) independently coded the co-design session transcripts. An analysis meeting was held including the final author to discuss codes and potential subthemes. Following this meeting, CB collated all codes into a common set of subthemes and shared the collated analysis with the peer researchers for validation. Results of Phases 1 and 2 are presented together. For Phase 3, findings were initially collated by the lead facilitator of the ToC workshop (SH). The ToC model was refined by CB following further analysis using the verbatim transcript of the workshop and participant email feedback. The ToC was designed to combine the purported outcomes of the hope-focused intervention (i.e., critical realism's actual tendencies [56]) with

structural influences that could impact these outcomes (i.e., generative mechanisms [56]). Findings from all three phases, plus relevant evidence from prior research and PCIE activities, were then integrated (Table 3). This tabulated synthesis was used to finalise the intervention blueprint (Table 4).

With respect to reflexivity, our analysis was influenced by the fact that our team comprises mainly White-identifying females. This matches the majority of the participants across all project phases. Many of our research team originate from Sussex or Kent, with all authors having lived and/or worked in at least one of these regions. All team members were thus familiar with growing up in coastal communities and/or inhabiting/working in these regions later in adult life. Multiple members of the team have experience of being NEET in adolescence or early adulthood, including whilst experiencing mental health problems, and in supporting NEET family members. The team additionally includes public health and mental health practitioners, with practice-based experiences of delivering psychosocial interventions to young women with social and mental health problems. Our approach to study design and analysis was influenced by these experiences, and the frequent degree with which the narratives of participants evoked personal and professional resonance.

## Results

### Findings from Phase 1 (understanding population needs and intervention requirements) and Phase 2 (co-creating intervention parameters)

Integrated findings from Phases 1 and 2 are presented in Table 3 and summarised narratively below. More detailed results, including illustrative quotes, are presented in S1 and S2 Tables. Co-design participant rankings of proposed intervention modules, activities, and outcomes are presented in S1–S3 Figs. A video summarising key results of Phases 1 and 2, with a particular focus on the narratives of NEET young women is additionally available: @Hopeinhealthresearch, https://www.youtube.com/watch?v=d03W9ax52QQ&t=10s.

**WHY (intervention rationale, theory, and goal).**   Results from the Phase 1 qualitative interviews and Phase 2 co-design sessions suggested that the cognitive model of hope aligned with participant experiences. When asked how they understood the word 'hope/hopefulness', Phase 1 participants from all groups spoke about hope as future-oriented, primarily cognitive in nature, and described it in relation to specific (usually functional) goals. Participants did additionally identify emotional components to hope, i.e., with respect to the intrapsychic experience of being hopeful and observable manifestations of hopefulness in other people. Young women (Phases 1 and 2) and practitioners (Phase 1) emphasised the importance of making the focus of the intervention on hope very explicit. They emphasised that this is less stigmatising than a diagnostically driven mental health intervention, and provides greater potential to prevent and ameliorate a spectrum of mental health problems. They suggested that prioritising EET engagement is not necessarily attractive nor helpful as an intervention focus. However, they acknowledged that many NEET young women would want to engage in EET activities and have goals aligned to these areas, thus all stakeholder groups (Phase 1) suggested that intervention providers should be equipped to offer activities for developing skills and knowledge for scaffolding EET (re)engagement. There was agreement across Phases 1 and 2 about the likely benefits of a structured, manualised, and modular programme to teach specific skills that are relevant to goal setting and pursuit as building blocks of hope. In Phase 2, NEET young women emphasised the potential for broad transferability of skills related to hope. All three groups (Phases 1 and 2) identified likely secondary outcomes to increasing hope of goal attainment, better functioning, and improved mental health and/or reduced mental health symptoms.

**WHAT (materials and procedures).** Phase 1 and 2 interviewees agreed that intervention materials provided to NEET young women should be accessible, varied in style (e.g., paper-based and online), and used flexibly to suit different needs, preferences, and styles of learning. Concern was raised by young people and practitioners about materials being written using patronising or overly positive language that might seem glib or invalidating. With respect to procedures, all stakeholder groups (Phase 1) agreed that the intervention should be inclusive to as many NEET young women as possible, with late adolescence through to early adulthood as a relevant staging period due to the critical developmental transitions and challenges occurring in this time. There was broad agreement in Phases 1 and 2 about the value of organising behavioural, cognitive, and interpersonal content into structured modules. Goal-directed behaviour change was identified by all groups as a key focus after initial engagement work and psychoeducation. This would include helping young women to identify meaningful goals, break them down into small steps and clarify specific goal pathways, explore and mitigate barriers, re-select goals when needed, and reflect on progress. In Phase 1, gradual and responsive pacing was emphasised as crucial to avoid overwhelming participants who may be ambivalent about change and severely lacking in confidence. In both phases, explicit discussions about the nature of hope and its enhancement were advised to begin after initial work to develop a more positive sense of self, e.g., through identifying interests and strengths. As part of the process of orienting participants to the concept of hope, practitioners (Phase 1) emphasised the importance of personalising psychoeducation by exploring the unique meaning of hope to each young woman, their historical experiences of hope, and factors affecting current hopefulness. All three stakeholder groups (Phases 1 and 2) additionally recognised that NEET young women often have impoverished social worlds, and that interpersonal content would usefully address ways to increase social support and connectedness. This would include helping young women to identify and strengthen existing positive relationships within their social networks, as well as expanding networks by actively creating opportunities for new connections.

Phase 2 participants endorsed six proposed modules, each organised around a theme relevant to the enhancement and maintenance of hope, that would incorporate a mix of psychoeducational, behavioural, cognitive, and interpersonal content (see Table 4). There was agreement that the first module should focus on introducing the intervention model, building rapport, generating a more positive sense of self, sparking interest in structured activities, and enhancing mood. The next module was intended to introduce the concept of hope in greater depth (Module 2). This would then be followed by successive modules to develop relevant skills and enhance resources to support ongoing hopefulness: visualising a positive future and clarifying motivational values (Module 3); identify and work towards personally meaningful goals (4); increase social support for hope (5); and mitigate challenges and barriers to maintaining hope (6). Modules 1 and 3 were ranked as most important (S2 Table and S1 Fig), but there was support for all modules and their respective foci.

Although not ranked higher than other components, co-design participants appreciated psychoeducation (ideally presented using short videos) as necessary to introduce concepts and scaffold learning practicable skills. Highly rated module activities included: identifying interests, meaningful activities, and values; activity scheduling, identifying hopeful life periods and domains and sources of hope; guided goal visualisation and goal steps planning; and hopeful social network mapping (see S2 Table and S2 Fig). Co-design participants also advocated the inclusion of lived experience perspectives as part of intervention materials and providing resources that the participants could use to explain the intervention approach and their progress to others around them. The participants suggested that the intervention could include components focused on EET activities and/or mental health problems as individually desired.

Participants identified the most important outcomes as hope (general and EET domain-specific), wellbeing, help-seeking intention, social relationships, and time spent in EET and other meaningful activities (S2 Table and S3 Fig). Participants identified that improving mental health and/or reducing risk of mental health problems is highly important but should not be the primary outcome.

**WHO (intervention provider).** All three stakeholder groups (Phase 1 and 2) agreed that the intervention would be more effective and engaging if prospective participants were supported through the process by a trusted individual who was empathic, validating, non-judgemental, hopeful, encouraging and consistent. A supporter's ability to form positive relationships was viewed as being more important than any particular professional training or qualifications. Training and qualifications were considered by practitioners (Phase 1) to potentially discourage NEET young women from sharing their experiences and engaging in support. Interpersonal continuity was strongly emphasised, and young people described a litany of experiences in which supportive interventions had ended abruptly, typically because of operational restrictions on longer-term support or staff turnover. NEET young women (Phase 1 and 2) and relatives (Phase 1) emphasised that NEET young women will vary with respect to the type of person they would want to support them with a hope-focused intervention; for some a peer would be preferred, but others would value someone with whom they have a professional relationship.

Practitioners (Phase 1) identified resource limitations to do with skilling-up existing staff from over-stretched statutory and community agencies and emphasised that the intervention's reach and scalability could be aided by participation of trained lay community members. Relatives and practitioners (Phase 1) both suggested that engagement and continuity might be strongest if prospective intervention providers/supporters were already personally known to NEET young women, such as foster carers and existing residential support workers, rather than introducing a new cohort of specialist practitioners. Embedding the intervention in the community was also seen as critical to sustainability. NEET young women and practitioners (Phase 1) identified the importance of intervention providers being trained to deliver the intervention using a person-based and flexible approach. Practitioners emphasised the need for training in the intervention approach, but discouraged it being delivered by providers with extensive psychotherapy training as this may undermine supporters' receptiveness to focusing on a novel approach to enhancing hope. All stakeholder groups (Phase 1) recommended that the intervention should be overseen/supervised by professional experts in youth work, even if the direct providers were non-specialists in the community.

Scenarios generated from Phase 1 were explored in greater detail in Phase 2. Particular traction was found with a 'youth-initiated mentoring' (YiM) model [60, 61] in which young people would identify a trusted individual with whom they had a pre-existing relationship and who they would like to support them through the intervention. Co-design participants responded favourably to YiM, with two particular caveats. First, alternative mentors would need to be provided for young women who could not identify a mentor themselves. Second, careful attention (and training) would be needed around negotiating boundaries and ensuring confidentiality when introducing the intervention into a pre-existing/ongoing relationship.

**HOW (mode of delivery).** Overall, a primarily one-to-one intervention was seen to be most accessible and attractive to the majority of NEET young women in Phases 1 and 2; offering a greater sense of privacy and safety and preserving a focus on individual need. However, the flexible use of small group-based activities, as part of the intervention and/or planned as part of activity scheduling more broadly, was also endorsed. Practitioners (Phase 1) additionally emphasised the benefits of including at least some self-directed components to make the intervention more accessible and create a space for NEET young women's empowerment.

There was additionally agreement in Phases 1 and 2 that sessions should ideally be in-person, but that flexibility would be needed to engage young women according to their dynamic preferences, offering remote voice or video-call sessions if preferred, for example, in the context of a period of heightened anxiety.

**WHERE (location).** NEET young women and practitioners (Phase 1) emphasised that the hope-focused intervention should fit within the existing youth support system. They stated that individuals and services with whom NEET young women might come into contact should be aware of the intervention and able to support young women to access it. All three stakeholder groups (Phase 1) advised that the intervention should be easy and quick to access, without long waiting periods. Waiting lists themselves were seen to undermine hope and reaffirm and reinforce NEET young women's feelings of hopelessness and worthlessness. All three stakeholder groups (Phase 1) agreed that information about the intervention should be disseminated clearly and accessibly online. It was also unanimously agreed (Phase 1) that in-person sessions should take place in accessible and non-stigmatising community locations with good transport links. Phase 1 practitioners and Phase 2 participants suggested that sessions could helpfully include meetings outdoors.

**WHEN and HOW MUCH (sessions, schedule, duration, intensity, and dose).** In Phase 1, family members emphasised the importance of an intervention that involves regular sessions, so that young women feel it is something consistent and reliable. Practitioners (Phase 1) suggested that the intervention should begin with more frequent sessions, then reducing to a lesser intensity before finishing. Both practitioners and NEET young women in Phases 1 and 2 emphasised the importance of young women not feeling rushed to complete the intervention. Moreover, these participants additionally emphasised that all aspects of delivery (session number and duration, timing, pacing) should be flexible according to individual participants' preferences. Phase 2 participants additionally recommended that activities which could be used beyond the formal end of the intervention would be beneficial.

**TAILORING (personalisation).** Personalisation according to NEET young women's goals, interests, preferences, and needs was espoused by all stakeholder groups in Phases 1 and 2. All stakeholder groups agreed that customisable and adaptable intervention activities would be essential to intervention engagement. Phase 2 participants recommended the provision of a menu of adaptable activities that could be completed in different formats such as writing, discussions, visualisations, and using creative arts. Co-design participants encouraged an early discussion as part of the intervention to determine session focus. Subsequently, activities should be selected and sequenced flexibly according to individual preferences and needs, building in ongoing opportunities for reviewing and returning to activities from earlier modules.

## Phase 3 findings: Theory of change (ToC)

The ToC (Table 2) sets out the inputs, participants, and assumptions underlying the hope-focused intervention and details the specific components and their links to purported outcomes. Key conclusions from the ToC workshop are integrated with all other evidentiary sources in Table 3.

Phase 3 participants agreed with the utility of first focusing on time spent in meaningful activity and increasing positive sense of self, before moving on to explicit discussions about hope. Alongside this, the group stated that it might be challenging for young women to engage in the intervention if basic needs, for example, safe housing and sleep, were not met. Nonetheless, participants acknowledged that this might be their agenda as practitioners and that young women do not necessarily prioritise basic needs above their future goals. The group identified

**Table 2. Intervention theory of change.**

**HOPEFUL Theory of Change Model**

**Problem**: Young women who are Not in Education, Employment and Training (NEET) lack hope and are at risk of potentially long-term mental health problems and social exclusion

**Focus**: Enhancing hope through a structured psychosocial intervention, delivered using online and printed materials with 1:1 support from a youth-initiated mentor

| Inputs | Outputs | | | Outcomes | | | | |
|---|---|---|---|---|---|---|---|---|
| | Participation | Activities | Change in outcome category facilitated by (module) | Mechanisms of outcome | Category | Short-term (0 to 6 months) | Medium-term (>6 to 12 months) | Long-term (>12 months to 5 years) |
| **People and services**<br><br>• Mentors<br>• Mentor trainers/ supervisors<br>• Referrals initiated by relevant statutory and community and voluntary sector organisations and/or directly from NEET young women<br><br>**Resources and materials**<br><br>• Workbook (online/ paper) for young woman, presenting six modules; each focusing on a specific topic related to hope and containing a menu of activities<br>• Training resources, supervision, and manual for mentors | • Young women who are NEET and living in deprived coastal communities<br>• Youth-initiated mentor for each young woman (or alternative mentor identified if needed) | **HOPEFUL intervention delivery**<br><br>• 1:1 psychosocial support, guided by a mentor, according to a structured but flexible manualised programme<br>• Active encouragement to spend time outside of the home (e.g., walking conversations; behavioural activation)<br>• Completion of workbook by young person<br>• Group component possible, e.g., toward end of the intervention<br>• Basic needs identified and addressed<br>• Training and supervision<br>• Provided to mentors using scalable formats (e.g., video-based training; group supervision) that and proportionate to intervention requirements | • Hopeful mentoring relationship (1–6)<br>• Behavioural activation (1–6)<br>• Learning about hope and its sources (2)<br>• Imagining positive future self (3)<br>• Practising skills in goal setting and pursuit (4)<br>• Increasing positive social relationships (5) | • Personal goal attainment<br>• Quality of mentoring relationship | **Hope** | • Improved motivation<br>• Recognise the importance of setting achievable goals<br>• Improved confidence to change behaviour<br>• Ability to set realistic goals and develop plans | • Raised life aspirations | • Sustained increase in hope<br>• Sustained raised life aspirations |
| | | | • Boosting time spent in activities outside the home (1–6)<br>• Addressing basic needs, e.g., sleep (1)<br>• Identifying interests and strengths (1)<br>• Change in mental health and well-being is additionally facilitated by the outcome of increased hope | | **Mental health and wellbeing** | • Increased self-awareness and ability to reflect<br>• Improved sense of identity | • Reduced anxiety<br>• Reduced depression<br>• Reduced suicidality<br>• Decreased mental health support needs | • Improved resilience<br>• Reduced alcohol and substance misuse<br>• Reduced contact with youth custody/ criminal justice orders<br>• Decreased mental health service use, including A&E |
| | | | • Hopeful mentoring relationship (1–6)<br>• Identifying interests, strengths and values (1, 3)<br>• Imagining positive future self (3)<br>• Practising skills in goal setting and pursuit (4)<br>• Increasing social capital via social relationships (5)<br>• Change in EET is additionally facilitated by increased hope | | **Education, employment, and training (EET)** | • Improved capability (skills and interests) | • Increased awareness of careers and access routes<br>• Entered EET | • Maintaining meaningful EET<br>• Motivation to upskill, seek promotions and continue to develop |
| | | | • Positive and consistent mentoring relationship, and any group activities (1–6)<br>• Practising communication skills (5)<br>• Understanding characteristics of hopeful relationships (5)<br>• Change in social functioning additionally facilitated by increased hope | | **Social functioning** | • Improved social skills<br>• Increased opportunities to form meaningful friendships<br>• Increased levels of trust in social relationships<br>• Expanded social networks and support systems | • Increased network of support and positive relationships with other people | •Improvement in parenting skills |

*(Continued)*

**Table 2.** (Continued)

| | | | | | | | |
|---|---|---|---|---|---|---|---|
| | | • Supportive relational environment through mentor (1–6)<br>• Increasing positive social relationships (5)<br>• Practising communication skills (5)<br>• Change in help-seeking outcomes additionally facilitated by increased hope | | **Help-seeking** | • Improved knowledge of available support and how to access it | • Confidence to seek and ask for the right type of support | • Sustained confidence to seek and ask for the right type of support |

**Assumptions**: Capacity in existing services to train/supervise mentors; HOPEFUL intervention is viewed by stakeholders as an acceptable and feasible intervention; Intervention dose = 4 sessions including at least one session each from modules 1, 2, and 4.

that communication skills are often an issue for young women and that the intervention could and should help improve these. Participants identified that using tools to capture outcomes is important to understand the impact of the intervention, but that this should also be done as part of the intervention itself, e.g., using metrics to discuss progress with young women, to enable productive conversations about how and why changes have (or have not) occurred. Finally, participants supported the use of the YiM model believing that this would help in the context of the lack of providers in coastal areas. Moreover, they emphasised that there would be particular benefit to mentors who had greater shared experience with NEET young women and could help to reduce social isolation among mentees.

## Integrated findings and intervention blueprint

Phase 1–3 findings were integrated alongside PCIE and scientific evidence in Table 3. These findings were used to refine the draft intervention blueprint, with the finalised version presented in Table 4. The intervention is named 'HOPEFUL'. The blueprint describes the key intervention parameters and includes the focus and content of six discrete yet interconnected modules. For each module, the blueprint articulates key concepts (psychoeducation components) and 'core' activities that are considered necessary to ensure that the intervention can be delivered (for example, discussing the intervention model and agreeing how the mentoring relationship will work in Module 1, and reviewing intervention progress in Module 6). The blueprint then identifies a menu of selectable activities, which can be adapted for delivery in different formats such as discussions and creative arts. The activities aim to facilitate the application of concepts learned about in the psychoeducation portion. Finally, optional 'takeaway' (i.e., homework) activities are provided, providing opportunities to practice the intervention techniques in a self-directed fashion and/or to involve others from the young women's wider social networks, e.g., sharing their personal goals with supportive others. It is recognised that modules may require more than one meeting with a YiM, although some young women may prefer to complete intervention activities (or modules) in a more self-directed fashion overall. With this in mind, the intervention blueprint provides guidance for a range of 4–12 sessions, approximately hourly in length and held weekly, but emphasises collaborative flexibility in determining session number, duration, and pacing.

**Table 3. Integrated findings.**

| Research Question | TiDiER item | | Evidence Source | | | | |
|---|---|---|---|---|---|---|---|
| | | | Literature | Phase 1 Stakeholder Interviews | Phase 2 Co-design Sessions | Phase 3 Theory of Change workshop | Additional Public and Community Involvement and Engagement (PCIE) activities |
| i: What format and setting(s) are viable for delivering a brief, low-cost, hope-focused intervention to NEET young women aged 16–24 years in deprived coastal communities? | WHY | | Psychological intervention explicitly focused on learning to be more hopeful | • Hope theory identifies hope as a particular way of thinking that is amenable to change [33]<br>• Hope as a robust predictor of positive youth outcomes [71–73], buffer against the impact of negative events [74], and transdiagnostic mechanism of psychotherapeutic change [75]<br>• Particular dearth of theory-driven intervention for the NEET population [17]<br>• NEET young people, especially females, tend to exhibit low hope and often inhabit social spaces characterised by hopelessness and pessimism [4, 30, 39, 41] | • All interview participants agreed on hope as an important and relevant intervention focus for NEET young women with respect to their individual needs and experiences, and constellations of complex vulnerabilities including systemic hopelessness<br>• All interview participants agreed that the intervention should explicitly target hope<br>• Practitioners emphasised that a hope-focused intervention should have transdiagnostic relevance, and would likely be accessible and non-stigmatising | • All co-design session participants agreed that the intervention should explicitly target hope, and not be framed as explicitly targeting EET activities<br>• Co-design participants emphasised the usefulness of being able to learn how to apply the skills of hope | • Practitioners supported the explicit focus on hope | • Youth PCIE group emphasised importance of framing hope as a changeable "mindset"<br>• Parent PCIE consultant commended focus on hope and termed it an important message to offer to young women, emphasising support to live the best life they can in the moment as opposed to focusing on mental health recovery or EET activities |
| | WHAT | | | | | | |
| | | Materials | Manualised intervention, structured with a workbook offered both online and on paper | • Guided self-help using printed/online materials may be equally as effective as more specialised face-to-face psychological intervention [76] | • Young women and practitioners agreed on a workbook to support intervention delivery, with the option of using printed and/or online materials | • Co-design participants agreed that there should be an accessible, non-patronising, workbook that is available online and on paper<br>• Co-design participants encouraged the use of non-value-laden language, the inclusion of video materials and the incorporation of lived experience perspectives | • Support for a manualised and structured intervention with a workbook | • Youth PCIE group recommended considering accessibility of materials for people with different abilities, for example, recording instructions as voice summaries (online workbook version) |
| | | | Guidance manual for intervention supporter | • Practitioners who experience greater role legitimacy and adequacy form more positive therapeutic relationships with vulnerable young people [77] | • Practitioners emphasised the need for intervention providers to agree common ways of discussing and enhancing hope | | • Support for the provision of a manual for intervention supporters | • Youth PCIE group recommended producing manual for intervention supporters that emphasises helpful language |

*(Continued)*

**Table 3.** (Continued)

| Research Question | TiDiER item | | Evidence Source | | | | |
|---|---|---|---|---|---|---|---|
| | | | Literature | Phase 1 Stakeholder Interviews | Phase 2 Co-design Sessions | Phase 3 Theory of Change workshop | Additional Public and Community Involvement and Engagement (PCIE) activities |
| | Procedures | Inclusive access for 16-25-year-old NEET young women, especially at transition points | • Emerging adulthood (e.g., 18–25 years) as a period characterised by identity exploration, instability, self-focus, feeling in-between, and (potential to) experience a lot of possibilities [78]<br>• The most severe and socially-disabling mental health problems begin before age 25 years [79, 80], and socio-occupational withdrawal is a key robust risk factor of transition from subthreshold difficulties to enduring diagnosable disorders [7, 8]<br>• Hopes seems most closely linked to positive social outcomes for adolescents compared to adults [81]<br>• Young people emphasise the importance of supporting the hope of marginalised groups [42] | • All three stakeholder groups agreed on late adolescence to early adulthood as the right period within which a hope-focused intervention should be offered<br>• All three stakeholder groups identified social and occupational transitions as key challenges for this age group, and identified these points as relevant for intervention delivery | • Co-design participants emphasised that the intervention should be acceptable and accessible to young women of diverse backgrounds, and at times of life transition | • Practitioners emphasised that it is difficult for NEET young women to work towards achieving other outcomes unless their basic needs are met and thus these needs should be addressed before commencing work focused on hopes and goal pursuit | • Youth PCIE emphasised importance of ensuring accessibility for marginalised young women, e.g., considering neurodiversity, gender identity and sexual orientation<br>• Parent PCIE consultant emphasised that access to hope-focused intervention should not be diagnosis-led |
| | | Gentle introduction to focus on hope | • Hope should first be built implicitly through positive encouragement and hopeful language, with more explicit discussions and activities to follow [42]<br>• Intervention supporters can gently and implicitly build hope early on by acting as a 'cheerleader', providing encouragement, and modelling positive self-talk [82] | • Practitioners emphasised that a trusting relationship with supporter, and work to enhance positive sense of self, should occur before introducing explicit focus on hope | • Initial module to scaffold mentor relationship and increase positive sense of self, before modules focused explicitly on hope | • Support for building sense of safety and positive self before beginning to work on hope | • Support for a gentle introduction to hope |

*(Continued)*

**Table 3.** (*Continued*)

| Research Question | TiDiER item | | Evidence Source | | | | |
|---|---|---|---|---|---|---|---|
| | | | Literature | Phase 1 Stakeholder Interviews | Phase 2 Co-design Sessions | Phase 3 Theory of Change workshop | Additional Public and Community Involvement and Engagement (PCIE) activities |
| | | Behavioural, cognitive, interpersonal, and additional intervention components | • Time spent in structured activity, social relationships and mental health are connected for NEET young people [69]<br>• Young people with complex mental health problems emphasise the importance of a sense of belonging, meaningful ways to spend their time, and achieving small goals to facilitate hopeful thinking [83]<br>• Interventions involving behavioural activation, with a focus on personally meaningful activity, result in young people feeling more hopeful [42, 84–87]<br>• Mapping one's values can help to identify long-term ambitions and priorities; motivating the individual to take immediate steps towards longer-term goals [82]<br>• Confidence-enhancing activities benefit NEET young people [88]<br>• Enhancing social identities is a means of improving agency, meaning, and health for vulnerable groups [89, 90] | • All three stakeholder groups agreed on the usefulness of NEET young women being supported to identify and engage in meaningful activity<br>• Parents and practitioners encouraged the use of psychoeducation and cognitive strategies such as strengths-spotting<br>• All three stakeholder groups emphasised the importance of building social connectedness, with young women and practitioners emphasising the value of social networks around NEET young women becoming more hopeful and encouraging<br>• All three groups additionally identified that the intervention should be able to offer components focused on supporting engagement in EET activities, and young women recommended the intervention be able to help support skills for coping with mental health challenges | • Co-design participants emphasised the usefulness of components that are behavioural (increasing meaningful activity, goal visualisation, setting and pursuit, mitigating goal barriers), cognitive (defining hope, exploring influences on and sources of hope, strengths-spotting, building self-esteem), interpersonal (exploring characteristics of hopeful relationships, reflecting on social support and how to increase it, sharing details of the intervention and goals with others), and additional (support for EET engagement and managing mental health/problems, at individual preference) | • Practitioners encouraged intervention components designed to improve sense of positive identity, confidence, emotional resilience, social networks, communication skills, and to raise aspirations, track change as it happens, and reduce social isolation and the use of unhealthy coping strategies<br>• Practitioners emphasised that spending time in meaningful activity can help generate aspirations for the future<br>• Practitioners recommended components focused on building life skills, for example, budgeting<br>• Practitioners recommended explicitly capturing, and then discussing, progress during the intervention is productive | • Parent PCIE advisor emphasised benefit of supporter helping young women to access activities |

(*Continued*)

**Table 3.** (Continued)

| Research Question | TiDiER item | | Evidence Source | | | | |
|---|---|---|---|---|---|---|---|
| | | | Literature | Phase 1 Stakeholder Interviews | Phase 2 Co-design Sessions | Phase 3 Theory of Change workshop | Additional Public and Community Involvement and Engagement (PCIE) activities |
| | | Importance of non-specific factors in intervention support | • Effective hope-focused intervention occurs within a positive interpersonal context [42]<br>• Positive professional expectations of what young and/or vulnerable people can achieve predict the formation of positive therapeutic relationships, and better social and occupational outcomes [91, 92]<br>• Clear expectations, collaborative practice, and taking to build therapeutic rapport based on meaningful connection are essential to working with young people with complex problems, including social and occupational withdrawal [82] | • All three stakeholder groups emphasised the importance of interpersonal continuity with respect to intervention support<br>• All three professional groups emphasised the importance of the intervention supporter being hopeful, validating, authentic, reliable, non-judgement | • Co-design participants emphasised the importance of the bond with the supporter, and the need for them to listen and validate<br>• Co-design participants recommended a very collaborative approach, in which the supporter also completed activities | • Agreement regarding the important of a consistent and supportive relational environment | • Support for a supportive environment around intervention delivery |
| | HOW | Primarily one-to-one intervention with group components at preference | • Individual, group, and mixed hope-based interventions can have positive effects on hopefulness and other positive youth outcomes [42]<br>• Engagement of NEET young people is scaffolded by providing individual support for young people who experience multiple barriers to re-engagement and/or have been NEET long-term [43] | • All three participant groups emphasised that one-to-one delivery can be the most suitable mode for many NEET young women, although young women and practitioners emphasised too the benefits of group-based approaches<br>• Young women and practitioners encouraged the opportunity for NEET young women to be able to access the intervention one-to-one and/or in group sessions according to their individual preferences | • Co-design participants emphasised the importance of there being options for one-to-one and group-based components | • Practitioners agreed primarily one-to-one delivery, with group-based approaches as relevant | • Youth PCIE group recommended group sessions/ activities arranged according to educational stage |

*(Continued)*

**Table 3.** (Continued)

| Research Question | TiDiER item | | Evidence Source | | | | |
|---|---|---|---|---|---|---|---|
| | | | Literature | Phase 1 Stakeholder Interviews | Phase 2 Co-design Sessions | Phase 3 Theory of Change workshop | Additional Public and Community Involvement and Engagement (PCIE) activities |
| | | Supported and self-delivered elements | • The use of self-help materials can be nearly as effective as supported intervention [93] and may be well aligned with adolescent and young adult drives for independence and autonomy [94, 95] | • All three participant groups emphasised the potential benefits of someone supporting young women to engage in the intervention, particularly with respect to the benefits of talking about goals with someone supportive and encouraging<br>• Practitioners emphasised that the inclusion of some self-directed components could make a space for NEET young women to feel empowered and develop more skills | • Co-design participants emphasised the importance of there being a balance between supported/ group-based and self-directed components<br>• The benefit of activities that could be continued to be used after the end of the intervention was emphasised | | |
| | WHERE | Accessible via self- and professional-referral, either as a standalone intervention or across the system | • Hope-based interventions can enhance hopefulness and other positive youth outcomes when accessed in health, education, and community settings [42]<br>• Complex access pathways can undermine young people's engagement and satisfaction [29, 96, 97] | • All three participant groups recommended the intervention be easy to access via self-referral and professional referral, with provision or signposting across the system, including health (primary and secondary care), social, educational, community, youth services | • Co-design participants emphasised the need to advertise the intervention, using printed and digital media, in spaces attended by young women | • Practitioners identified existing NEET and youth support services as potential candidates for providing the intervention | |

*(Continued)*

**Table 3.** (Continued)

| Research Question | TiDiER item | | Evidence Source | | | | |
|---|---|---|---|---|---|---|---|
| | | | Literature | Phase 1 Stakeholder Interviews | Phase 2 Co-design Sessions | Phase 3 Theory of Change workshop | Additional Public and Community Involvement and Engagement (PCIE) activities |
| | | Primarily in-person sessions, offered in accessible community venues, including outdoors | • Delivering programmes in 'neutral' community settings, without the stigma or other negative connotations of statutory services, encourages NEET young people to engage [43]<br>• Robust evidence for the benefits of time spent in nature for mental and physical health and wellbeing [98–100]<br>• Hope-based interventions that involve natural settings can enhance hopefulness for young people [42] | • Parents and practitioners identified in-person sessions as offering young women a better experience, greater connectedness, and being aligned with young people's post-pandemic preferences for intervention delivery<br>• All three stakeholder groups emphasised that the location of each session should be flexible according to young women's preferences, and may need to include online and/or telephone delivery instead of face-to-face meetings<br>• All three stakeholder groups emphasised the importance of in-person sessions being in locations that were non-stigmatising, local, and with good transport links<br>• Practitioners encouraged holding intervention sessions outdoors when appropriate to do so | • Co-design participants supported primary delivery of the intervention in person, in accessible and non-stigmatising community spaces<br>• Co-design participants supported an aim for 'active' delivery, including sessions conducted outdoors | • Practitioners emphasised youth centres as potentially appropriate, safe-seeming spaces for intervention sessions<br>• Practitioners emphasised the benefits of encouraging NEET young people to engage in activities outdoors and take sessions outside of clinical spaces | • Parent PCIE consultant emphasised the benefit of holding intervention sessions outdoors when possible |

(*Continued*)

**Table 3.** (Continued)

| Research Question | TiDiER item | | Evidence Source | | | | |
|---|---|---|---|---|---|---|---|
| | | | Literature | Phase 1 Stakeholder Interviews | Phase 2 Co-design Sessions | Phase 3 Theory of Change workshop | Additional Public and Community Involvement and Engagement (PCIE) activities |
| | WHEN and HOW MUCH | Modular structure delivered at pace determined in consultation with young person | • Even single session hope-based interventions can have positive effects on hopefulness and other positive youth outcomes [42] | • Young women and practitioners identified that offering the intervention in small "chunks" is helpful <br><br> • Young women and practitioners emphasised the importance of flexibility according to young women's preferences with respect to the pacing of therapy sessions and the overall duration, and that the intervention should not be too brief or end abruptly <br><br> • Parents emphasised that there should be the option to offer regular sessions | • Co-design participants encouraged a modular structure, recommending a six-module intervention, with flexible session number, pacing and duration | • Practitioners supported the use of a modular structure | • Youth PCIE group emphasised that more sessions is better, and supported the ability for duration to be determined by mentor and young person |
| | TAILORING | Person-centred approach with adaptable activities | • Young people want support services which offer flexibility [101] <br><br> • Engagement of NEET young people is increased by offering flexible support with a high degree of personalisation [43] <br><br> • Allowing for periods of disengagement and missed appointments is often necessary when working with young people with complex problems, including social and occupational withdrawal [82] <br><br> • Emerging adults find it helpful to receive encouraging and supportive messages and advice, especially when these messages relate to specific strengths and are aligned to specific goals [102] | • All three participant groups emphasised that hope is unique, and that the intervention should be person-centred, focusing on young women's individual goals and preferences <br><br> • All three stakeholder groups emphasised that intervention activities should be adaptable in order to be suitable for young women with different learning needs and engaging based on individual interests, including creative arts-based activities | • Co-design participants recommended the intervention begin with a discussion to establish the young woman's priorities, with all activities including homework being optional and adaptable (e.g., written, audio, visual/arts-based), and opportunities to review, revise, and repeat activities as desired | • Practitioners emphasised the utility of having a menu of different intervention materials and techniques (including written, outdoor, meditation, role play) to offer young people depending on their specific needs and preferences | • Youth PCIE group emphasised the importance of adaptable activities for people with different learning needs and preferences |

**Table 3.** (Continued)

| Research Question | TiDiER item | | Evidence Source | | | | |
|---|---|---|---|---|---|---|---|
| | | | Literature | Phase 1 Stakeholder Interviews | Phase 2 Co-design Sessions | Phase 3 Theory of Change workshop | Additional Public and Community Involvement and Engagement (PCIE) activities |
| iii. Who are possible non-specialists that could credibly deliver the intervention? | WHO PROVIDES | Supported by a youth-initiated mentor (or alternative when not available or preferred) with relevant experience yet who is not necessarily an expert | • Engagement of NEET young people in interventions is strengthened by non-hierarchical relationships between NEET young people and providers, and involving community members and/or peers in the delivery of programmes [43]<br>• Relationships with non-expert practitioners can support hope for young people with mental health and social vulnerabilities [91]<br>• More successful interventions for NEET young people have higher contact hours [17]<br>• Hope-based interventions can have positive effects when supported by non-experts/non-practitioners as well as when expert professional-delivered [42]<br>• Emerging adults, and their levels of hope, are best served by seeking advice and help from people with whom they have a pre-existing relationship characterised by high-quality support [102]<br>• Broad and significant benefits on health and functioning achieved using a youth-initiated mentor model [103] | • All stakeholder groups supported use of mentoring type approach, practitioners emphasised utility of NEET young women being able to pick a mentor that they already had a relationship with, either professional or personal<br>• All three stakeholder groups emphasised that the interpersonal approach of the mentor was more important than professional qualifications, although lived experience of being NEET, mental health knowledge, and knowledge and skills to support EET engagement were identified as beneficial characteristics<br>• Practitioners emphasised that involving non-experts as intervention supporters makes the intervention scalable and sustainable | • Co-design participants broadly supported the youth-initiated mentor model, but emphasised need for there to be alternative mentors for young women who could not identify, or preferred not to work with, someone they already had a relationship with<br>• Co-design participants recommended a supporter with experience of supporting young people, some mental health knowledge, and personal experience of being NEET, if possible | • Practitioners supported the use of a youth-initiated mentoring (with alternatives) approach, emphasising that NEET young people may be more likely to engage in a hope-focused intervention when it can be completed with the support of someone they already know and trust<br>• Practitioners stated that they thought it would work well to complete a specific hope-focused intervention within an existing mentoring or other type of support relationship with young people, if their service adopted this provision | • Youth PCIE group supported use of mentoring type approach and recommended that interested people could volunteer to be a mentor, as well as being invited to do so by a young woman |

*(Continued)*

**Table 3.** (Continued)

| Research Question | TiDiER item | | Evidence Source | | | | |
|---|---|---|---|---|---|---|---|
| | | | Literature | Phase 1 Stakeholder Interviews | Phase 2 Co-design Sessions | Phase 3 Theory of Change workshop | Additional Public and Community Involvement and Engagement (PCIE) activities |
| | | Digestible training on intervention approach, non-specific factors, and principles for safe and effective use | • Supporters' abilities to use non-specific skills, e.g., formation of positive therapeutic relationship, clarity regarding intervention approach, collaborative practice, flexibility and creativity, are essential to working with young people with complex problems, including social and occupational withdrawal [82] | • Young women and practitioners recommended intervention supporters have some training in interpersonal and communication skills and mental health literacy<br>• Practitioners recommended intervention supporters not have extensive training in any psychotherapeutic orientation | • Co-design participants identified core training topics as: intervention approach, especially focus on personal meaning; confidentiality; interpersonal and communication skills | • Practitioners emphasised willingness to be trained to support the use of the intervention training and recommended some basic training in working with a young person and trauma-focus | |
| | | Multi-professional supervisory input | • Regular and effective supervision is essential when supporting young people with complex mental health problems and co-present social disability [82]; hopelessness and isolation can vicariously affect therapists' own hope [104]<br>• It is helpful if the supervision team possess knowledge about local community resources and opportunities for EET and other activities [82]<br>• Young people who are socio-occupationally withdrawn who develop mental health problems have longer and more complex pathways to care, thus there is a need for all services and individuals supporting this group to help improve more rapid access when needed [29] | • Practitioners emphasised the need for interventionists to be provided with both emotional and technical support, with access to experts in mental health | | • Practitioners suggested support for intervention providers could be provided within the existing youth sector | • Youth PCIE group emphasised importance of young women being supported able to access mental health support alongside a hope-focused intervention |

(*Continued*)

**Table 3.** (Continued)

| Research Question | TiDiER item | | Evidence Source | | | | |
|---|---|---|---|---|---|---|---|
| | | | Literature | Phase 1 Stakeholder Interviews | Phase 2 Co-design Sessions | Phase 3 Theory of Change workshop | Additional Public and Community Involvement and Engagement (PCIE) activities |
| Outcomes | | | Hope as primary outcome, with wellbeing, help-seeking, mental health symptoms, and socio-occupational outcomes as secondary changes | • Evidence suggests that increased hope is associated with improved wellbeing, better daily, academic and vocational functioning, reduced depression, and suicidality among young people [72, 105–107] | • Young women and practitioners identified hope as impacting on EET activity engagement, daily functioning and social outcomes<br>• All three participant groups identified hope as likely improving mental health and/or reducing mental health symptoms | • Co-design participants emphasised that hope would likely be the main factor to change during the intervention, and that wellbeing, EET activities, social outcomes and mental health symptoms may also change but should not be the primary change target | • Practitioners identified help-seeking as an important secondary outcome<br>• Practitioners identified suicidality as an important secondary outcome, predicting that the intervention would decrease suicidality through increasing hope | • Young people emphasised that change in long-term outcomes might be relatively minor, but that it is plausible for changes to occur in social and occupational outcomes once hope has increased |

## Discussion

This paper presents the development of a hope-focused intervention (HOPEFUL) for young women who are NEET with a focus on UK deprived coastal community living, but potentially with wider applications to NEET and other vulnerable young people. The participatory process of developing the intervention blueprint used a phased qualitative design involving NEET young women, their family members, and statutory and non-statutory service providers as key stakeholders. We aimed to identify a viable format and setting for delivering a brief, low-cost intervention that could be credibly delivered by non-specialists. We also sought to identify meaningful short and long-term outcomes within an overarching Theory of Change. The resultant blueprint articulates a primarily one-to-one, in-person intervention, with content organised into six modules that aim to enhance hope by increasing positive self-regard and meaningful activity, identifying values and helping to visualise a positive future, facilitating the identification and pursuit of personally meaningful goals, and maximising social support for hope. Support for engaging with the intervention content is provided through a youth-initiated mentor (YiM) approach, with additional opportunities for self-directed activities.

All stakeholder groups considered hope of paramount importance, both as a universal process in good mental health and functioning, and of specific relevance to NEET young women living in deprivation, for whom intergenerational and socio-cultural transmissions of pessimism regarding life opportunities are common [39]. All groups agreed that the intervention should be explicitly framed around the concept of hope, for this offers an intuitive, engaging, accessible and non-stigmatising focus for the intervention. This contrasts with the primary emphasis on returning to EET within established NEET-specific programmes [17], but is in keeping with self-perceptions as an accessible intermediate intervention target to support movement towards (re)entry into EET activities [30]. All stakeholder groups intuitively defined hope in alignment with the cognitive model [33], i.e., in terms of goal-oriented future-focused cognition, and recognised the important ways in which hope can be developed and maintained in the context of positive relationships. These congruences attest to the relevance and utility of a mentor-supported intervention that explicitly targets hope.

**Table 4. Intervention blueprint.**

| TiDiER item | Intervention element | Element details |
|---|---|---|
| WHY | Psychological intervention explicitly targeting hope | • Intervention is explicitly framed as focusing on hope, with clear messaging that hopefulness is important in helping young people to live meaningful lives<br>• Drawing primarily on hope theory [33] to offer a psychosocial intervention that targets the enhancement of hope, through supporting NEET young women to increase their positive self-regard engagement in meaningful activity, and to enhance their skills in identifying and pursuing goals that are personally relevant<br>• Education, employment, and training (EET) should not be considered as the primary objective<br>• Changes in mental health outcomes are similarly not intended to be primary focus of the intervention but can be discussed in relation to link with positive, valued activity (e.g., overcoming negative cycles of social withdrawal/anxious avoidance/risk-taking behaviour that may provide temporary relief, but maintain negative affect, motivational difficulties, and maladaptive thoughts/behaviours in the longer-term)<br>• Supported by additional theories and principles of Social Recovery Therapy [82], brief behavioural activation [108], values [109] and self-affirmation theory [110], possible future selves [111], principle-guided approaches to transdiagnostic talking therapies [112] and 'wise' interventions [63]<br>• Supported by a Theory of Change co-produced with relevant stakeholders |
| WHAT | | |
| Materials | Workbook offered online and on paper | • Workbook used to structure the intervention and outline activities<br>• Able to print and/or save workbook for looking back on intervention once completed<br>• Inclusion of lived experience narratives of hope and goal pursuit<br>• Clear and accessible, yet non-patronising, language, and branding |
| | Companion manual for intervention mentors | • Guidance provided on implicit triggering of hopeful thinking through use of encouraging, yet not invalidating, communication<br>• Guidance on using terminology that is not overwhelming, for example, using words like "goals" rather than "success" and "achievement"<br>• Guidance provided on different ways of using intervention activities for young women with different learning and support needs, for example, considering neurodiversity<br>• Guidance provided on discussing the completion of 'takeaway' self-directed activities and identifying and mitigating barriers to doing so<br>• Guidance on needing to be understanding and creative in order to facilitate intervention engagement and be sensitive and responsive to potential disengagement<br>• Guidance provided on engaging in supervision and peer support networks<br>• Training materials contained/signposted for ongoing reference<br>• Guidance and resources provided on supplementary activities that do not form part of the core intervention package, but that are recommended for introduction when aligned with the preferences of recipients, e.g., resources on life skills such as budgeting, resources on managing mental health difficulties and mental health help-seeking, resources on preparing for studying, looking for work, preparing for interviews, and so on |
| Procedures | Inclusive entry criteria for NEET young women | Inclusive age range of late adolescence to early adulthood; 16 to 25 years<br><br>No specific mental health inclusion or exclusion criteria, but emphasis on young women being stable with respect to housing and not actively suicidal |
| | Gentle introduction of hope | • Structure of intervention designed to help develop rapport in mentoring relationship and support young person to enhance positive self-regard, before explicitly introducing a focus on hope<br>• Framing of hope as a mindset, a way of thinking, that is unique to every individual<br>• Framing of hope as changeable, i.e., that people can practice the skills and learn to be more hopeful |
| | Six modules involving psychoeducation, behavioural, cognitive, and interpersonal components | • Six discrete yet inter-connected modules, each focusing on a specific topic relating to hope<br>• Each module comprises a core psychoeducational component and menu of activities that can be selected<br>• Each module contains additional 'takeaway' activities that can be completed in a self-directed fashion between sessions to practice applying learning<br>• Modules 4–6 include explicit rating of attainment of personally-meaningful goals to enable identification of and reflection on progress<br>• Lived experience examples included (e.g., video narratives and quotes) throughout<br>• Each module contains session summary and discussion points developed for young women to share with people around them<br>• Modules: |

*(Continued)*

**Table 4.** (Continued)

| TiDiER item | Intervention element | Element details |
|---|---|---|
| | | 1. About me:<br>• Focus: i. introducing intervention and mentoring model, ii. introducing benefits of meaningful activity, ii. introducing benefits of focusing on character strengths<br>• Core components, e.g.: *Intervention overview and mentoring relationship agreement; Psychoeducation regarding benefits of positive self-regard and meaningful activity*<br>• Selectable session activities, e.g.: *Identifying interests and related activities; Strengths-spotting; Positive affirmations; Activity scheduling*<br>• Takeaway activities, e.g., *Activity diary*<br>2. About hope:<br>• Focus: learning about hope, why it matters, and exploring sources of hope<br>• Core components, e.g.: *Psychoeducation on the cognitive model of hope and its effects; Exploring own definition of hope*<br>• Selectable session activities, e.g.: *Self-report scales to identify domains of hope; Creating a hope timeline; Identifying sources of hope; Hopeful activity scheduling*<br>• Takeaway activities, e.g., *Recalling a hopeful memory*<br>3. My values:<br>• Focus: identifying values and considering the desired future<br>• Core components, e.g.: *Psychoeducation about values and their links to personally meaningful goals*<br>• Selectable session activities, e.g.: *Imagining hoped-for future self; Values mapping; Identifying and scheduling value-based activities;*<br>• Takeaway activities, e.g., *Values-based activity diary*<br>4. My goals:<br>• Focus: visualising, setting, and working towards goals<br>• Core components, e.g.: *Psychoeducation about goal setting and pursuit; Goal attainment scale*<br>• Selectable session activities, e.g.: *Guided goal visualisation; Goal sorting task; Breaking down goals into small steps*<br>• Takeaway activities, e.g., *Goal and values connections mapping*<br>5. My hope network:<br>• Focus: exploring how young person's social network supports hope<br>• Core components, e.g.: *Psychoeducation about relationships with other people and how they can affect hope; Goal attainment scale*<br>• Selectable session activities, e.g.: *Identifying characteristics of hope-enhancing relationships; Social network mapping; Communication skills*<br>• Takeaway activities, e.g., *Goal sharing log*<br>6. Staying hopeful:<br>• Focus: reviewing progress and planning how to maintain hope<br>• Core components, e.g.: *Psychoeducation about maintaining hope when encountering obstacles; Reviewing intervention progress, experience and accomplishments; Reviewing the mentoring relationship and agreeing arrangements for any ongoing contact; Goal attainment scale*<br>• Selectable session activities, e.g.: *Reviewing personally meaningful goals and identifying potential barriers and mitigations; Reviewing sources of hope and planning how to stay connected; Reviewing hopeful network and planning how to stay connected*<br>• Takeaway activities for ongoing self-directed use, e.g., *Write hopeful letter to future self* |
| | Primarily one-to-one in-person intervention | • Primarily a one-to-one intervention, in-person wherever possible, in which intervention tasks and activities are completed collaboratively<br>• Options for group components where possible and of interest to young women, e.g., gathering multiple young women engaged in the intervention together for group sessions facilitated by two intervention mentors |
| WHERE | Easily accessible across system | • Can be accessed and completed at no cost to young people<br>• Multiple access routes, e.g. provision/referral/signposting from NHS, social care, education, youth and community services, including emergency services |
| | Accessible in-person session locations | • Based in local community, in non-stigmatising spaces with good transport links<br>• Active sessions which encourage going outside of the home (e.g., walking conversations with the mentor; planning/scheduling activities for the young person to carry out in their own time in the community) |
| WHEN AND HOW MUCH | Flexible session pacing and duration | • Guidance provided (approximately 4–12 sessions of about an hour each, approximately weekly) but session number, pacing and duration determined collaboratively according to young person's preference, in discussion with the mentor |

*(Continued)*

**Table 4.** (Continued)

| TiDiER item | Intervention element | Element details |
|---|---|---|
| TAILORING | Person-centred approach | • Initial engagement should include a discussion between young person and mentor to establish preferences and boundaries guiding the intervention delivery<br>• Mentor should be consistent, offering continuity of support<br>• Mentor should be encouraging, hopeful, understanding, and non-patronising<br>• Sessions could be delivered in-person, online, and/or via telephone as needed on a sessional basis<br>• Hope should be as defined by the young person themselves, with the mentor providing opportunities to explore this<br>• Focus on activities, values, and goals as personally meaningful to the participating young person |
| | Menu of adaptable activities | • A menu of adaptable activities provided per module, that can be variably completed at participant preference using 1:1 ot group discussions, collaborative or self-guided creative arts, writing, outdoor activities, and/or self-study of printed materials/videos<br>• The number and nature of activities completed is decided according to young person's preferences, in discussion with the mentor<br>• "Takeaway" self-directed homework activities are provided, but are not mandatory |
| WHO PROVIDES | Youth-initiated mentor approach | • Supported by a "mentor," who is ideally youth-initiated (YiM) i.e., identified by the young person as someone with whom they have a pre-existing relationship and who they would like to be trained in order to support use of the HOPEFUL intervention, for example, a youth worker, football coach, teaching assistant, sibling or grandparent<br>• An alternative should be provided when a YiM cannot be identified, or is not preferred—the alternative should be drawn from the supervisory team, or another person from an organisation supporting the delivery of the intervention, or young women who has completed the intervention where safe and appropriate to do so |
| | Multi-perspective supervision | Scalable supervision methods, that require no additional practitioner resource, are emphasised<br><br>Supervision should be both expert and peer, requiring the creation of networks which contain both professional and lived experience experts alongside mentors using the intervention<br><br>Supervision networks will ideally contain experts with experience in youth, employment, education, social and mental health service provision, and ideally with people who have experience of being NEET<br><br>Supervision networks should aim to offer both emotional and technical support, with a focus on sustaining mentor hopefulness<br><br>Supervision networks should agree ground rules on how to offer support and discuss cases with respect to cross-organisation confidentiality and boundaries |
| | Digestible and scalable training package | • Training package that is digestible, accessible, and self-administered to ensure a wide variety of people can become mentors<br>• Training that focuses on understanding the intervention approach, non-specific relational aspects of support, specific procedural elements of module delivery, and being equipped to safely support young women to complete the package, including mentoring approach, confidentiality, boundaries, and signposting<br>• Additional training modules that focus on indicators that suggest signposting and/or referral for more specialist mental health support might be needed<br>• Additional that emphasises that, where necessary, possible, and appropriate to do so, mentors are encouraged to set-up additional opportunities for young women to engage in meaningful activity, including but not limited to EET-relevant activities, e.g., finding opportunities for social and hobby-based activities and accompanying a young woman to the first meeting, identifying educational courses or employment interviews and providing advocacy, CV review, and interview skills practice |

The proposed supportive role of a YiM is a particular innovation of the present study. This would be someone whom the NEET young person already knows (e.g., a relative, existing support worker, or sports coach) that they choose to support them with the intervention. In the Netherlands and USA where it originates, the YIM model has achieved impressive outcomes in improving academic and work functioning and physical and mental health in a variety of high-risk youth populations [61, 62]. The YiM model has strong promise for use with young people in low-resource settings from other countries, including in the UK. It requires fewer resources compared to traditional mentoring or professional intervention [61, 62], and uses existing assets to expand individual intervention benefits and propel them forward in time

[63]. Moreover, this model could help to address the issue found in the present study and in other research [30, 39], whereby NEET young people often experience projected hopelessness from their own families and wider communities. YiMs hold the potential to build support and social capital around NEET young women, increasing the resources available to them and helping them to maximise the extent to which they can draw on the support of caring adults [61], whilst engendering better access to relational hope and positive aspirations in their immediate surrounding networks.

The HOPEFUL intervention is in keeping with the 'wise' intervention approach [63]. Unlike traditional social reform, which targets either the individual or the environment, 'wise' interventions target how individuals engage with the wider system through augmenting basic psychological processes. HOPEFUL follows the five principles of wise interventions. First it is *psychologically precise*, by attempting to alter hope as a specific and intuitive form of meaning-making, i.e., encapsulating how agents engage with the social world. Second, it identifies *psychological processes as one factor in complicated causal systems*. This means that, for NEET young women to experience better health and functioning, the system around them must have the capacity for change. This may require other types of social reform, for example, NEET young women will stay NEET unless appropriate and accessible EET opportunities exist. Yet opportunity is not always taken up, and enhanced hope enables NEET young women to take advantage of this capacity where it does exist [42]. Third, HOPEFUL *encourages recursive change*, because what people believe about themselves and social situations readily becomes self-fulfilling and embedded in the structure of their lives [63, 64], and hope as a trait that young people and research evidence suggest is especially self-reinforcing over time [42]. Fourth, HOPEFUL has been developed with attention to *methodological rigour and process* and, fifth, takes into account *ethical considerations*. The intervention is based on robust psychological science. It tries to sustainably and accessibly embed hope-inspiring support in the system around young women to avoid contributing to NEET young women's perspective on being offered only short-term help with a high level of interpersonal turnover. Evaluation research should nonetheless now focus on the identification of any unexpected potential for harms.

As per other 'wise' interventions [63], HOPEFUL is designed to be minimally directive, intuitive and brief, and therefore, it is appropriate to position non-specialists in intervention support/delivery roles. This approach can be construed as an example of task-sharing, i.e., the deployment of non-specialists to expand the reach of statutory/community support provision. We envisage, for example, that local authority and voluntary sector practitioners could offer supervision to YiMs. The task-sharing approach has the potential to enhance implementation and scalability, because the YiM model means that HOPEFUL can be used at a population level, without a need for a new workforce to deliver it. Task-sharing approaches are not without implementation challenges, however, such as non-specialists lacking confidence in their competence, a lack of incentives to sustain involvement of non-specialists and other stakeholders beyond short-term research projects, and scepticism from specialist practitioners [65]. HOPEFUL, deployed using a YiM model, has the potential to mitigate at least some of these challenges due to the intuitive appeal of the hope focus and the established, trusting relationships on which the delivery approach depends. It is notable in the present study that professionals across a wide variety of health and other community services expressed a very high level of enthusiasm for the YiM model. HOPEFUL has been designed to be easy to understand and implement, to help build YiMs' confidence in their capabilities to support its use. Moreover, the use of the YiM approach should help to sustain the benefits of the intervention beyond the immediate lifetime of an evaluative research programme and provide a vital bridge between fragmented formal services and informal community support networks. Scalability

will be further enhanced by the creation of resource-efficient mentor training that requires little to no practitioner oversight or delivery; an approach used in other task-sharing initiatives, notably in Lower and Middle Income Countries (LMICs) [66]. These elements are reflected in the HOPEFUL intervention blueprint accordingly. The YiM approach may, nonetheless, create unique implementation challenges too. Therefore, attention needs to now turn to robust evaluation of intervention feasibility, effectiveness, and value for money. As hope is a basic and universal psychological process [67], learning from the evaluation of intervention effects and cost-effectiveness would be relevant to other vulnerable youth populations [42] and beyond.

In addition, important implications for the ongoing implementation of existing service provision arose in the present study. It was notable that NEET young women and their family members emphasised that simply the presence of available support was itself hope-inspiring. Where support provision occurs within the context of interpersonal continuity, young women feel more able to access it and more engaged in doing so. The importance in particular of feeling supported to work towards personally relevant goals was clear, as was the value of being able to do so at one's own pace without feeling rushed or that service provision might be abruptly discontinued. It was notable that discourses relating to sex and gender-based issues seemed largely absent from the narratives of NEET young women themselves. Nonetheless, professionals emphasised this group's particular need for provision that builds their hope, in the context of sex and gender-based inequities that maintain poor confidence and low life aspirations amongst young women.

## Limitations and strengths

Several limitations are important to note. We recruited fewer family members in Phase 1 and fewer NEET young women as Phase 2 co-design participants than intended. Difficulties in recruiting NEET young people have been reported in previous studies [68, 69] and, whilst we used multiple strategies to try to find and involve these participants, greater time to form relationships with more organisations and to promote the project more widely through social media may have resulted in greater participation of NEET young women and their relatives. We recognise that all NEET young women and their family members identified as White and spoke English as their first language. While minority ethnic groups make up a relatively small proportion of the population in the study localities, proactive efforts should be made in future studies to engage more ethnically diverse participants and explore any implications for intervention adaptations. Moreover, all the NEET young women interviewed in this study had completed GCSEs (or equivalent) or further study, and their needs and preferences may differ from NEET young people who have not completed such qualifications.

The involvement of people with lived experience in a number of roles was a strength of this study, albeit with challenges. Several recruited and appointed peer researchers withdrew for personal reasons. We were able to work flexibly nonetheless to engage people in peer researcher and other advisor and consultant roles, and hugely benefitted from working with these individuals. In particular, a peer researcher who was recruited fairly late in the project lifecycle played a central role in conducting and analysing the co-design sessions with NEET young women. We acknowledge that we presented co-design session participants with draft ideas for the intervention, rather than inviting ideas with no stimuli. However, we believe this to be an appropriate approach to co-design. Young people are well able to describe what they like and dislike about potential intervention components but may be less able to generate effective behaviour change techniques or design solutions themselves [44]. Indeed, when we asked young people in the Phase 1 qualitative interviews for ideas and preferences regarding a hope-focused intervention, we realised that participants found this to be a challenging task. We

consider that our approach, in which we showed a draft intervention outline and invited critical commentary, was a strength of the current project. We found that this way of working elicited directly relevant reflections and actionable suggestions from co-design participants, but also critical comments such that they appeared to feel able to challenge and disagree with our ideas. We acknowledge that involvement of family members and professionals in complementary co-design sessions would have enabled more in-depth consideration from their perspectives.

## Conclusion

The HOPEFUL intervention developed in this study offers a theory-driven practice innovation to prevent mental ill-health and improve social outcomes for a vulnerable and neglected group of NEET young women. By targeting a core ingredient that predicates goal-directed behaviour change and underpins well-being, this intervention has the potential to enhance hope as a means of helping NEET young women to thrive, and to augment how they interact with their surroundings. It has been designed with the potential for scalable implementation in mind, using simple and adaptable intervention materials and activities, and mobilising existing community members as intervention supporters rather than established practitioners. This intervention could complement other population-level initiatives that target wider structural challenges, such as poverty, isolation, lack of infrastructure, and gender-based violence and discrimination in under-resourced coastal areas and other areas of high disadvantage. Future research by our group will evaluate this intervention in practice.

## Supporting information

**S1 Table. Data analysis of Phase 1 stakeholder research interviews conducted with NEET young women, family members, and health and community practitioners.** Illustrative quotes are presented, labelled with participant number.
(DOCX)

**S2 Table. Data analysis of Phase 2 co-design sessions conducted with NEET young women.**
(DOCX)

**S1 Fig. Intervention module importance rankings by Phase 2 co-design participants.** Module names are presented with module number in parentheses. Primary axis presents percentage of participants ranking each module 1 (most important) to 6 (least important). Secondary axis presents mean rank.
(DOCX)

**S2 Fig. Intervention module activities importance rankings by Phase 2 co-design participants.** Module name and number presented as figure titles. Primary axes present percentage of participants ranking each module activity 1 (most important) to 4 (least important). Secondary axes present mean rank.
(DOCX)

**S3 Fig. Intervention outcomes importance rankings by Phase 2 co-design participants.** Primary axis presents percentage of participants ranking each outcome 1 (most important) to 10 (least important). Secondary axis presents mean ranks.
(DOCX)

## Acknowledgments

We are grateful to everyone who participated in this project. We wish to acknowledge the support for this study provided by the Youth Café and other members of the public who provided consultation input, additional project researchers based at the University of Sussex, and everyone who facilitated the involvement of participants in this project. We are grateful to the study steering committee who provided invaluable guidance in this work.

## Author Contributions

**Conceptualization:** Clio Berry, Julia Fountain, Lindsay Forbes, Daniel Michelson.

**Data curation:** Clio Berry, Leanne Bogen-Johnston, Abigail Thomson, Yelena Zylko, Alice Tunks, Sarah Hotham, Daniel Michelson.

**Formal analysis:** Clio Berry, Julia Fountain, Lindsay Forbes, Leanne Bogen-Johnston, Abigail Thomson, Yelena Zylko, Sarah Hotham, Daniel Michelson.

**Funding acquisition:** Clio Berry, Julia Fountain, Lindsay Forbes, Daniel Michelson.

**Investigation:** Clio Berry, Leanne Bogen-Johnston, Abigail Thomson, Yelena Zylko, Alice Tunks, Sarah Hotham, Daniel Michelson.

**Methodology:** Clio Berry, Julia Fountain, Lindsay Forbes, Abigail Thomson, Yelena Zylko, Sarah Hotham, Daniel Michelson.

**Project administration:** Clio Berry, Leanne Bogen-Johnston, Abigail Thomson, Alice Tunks, Daniel Michelson.

**Resources:** Clio Berry, Daniel Michelson.

**Supervision:** Clio Berry, Julia Fountain, Daniel Michelson.

**Visualization:** Clio Berry, Sarah Hotham.

**Writing – original draft:** Clio Berry, Julia Fountain, Sarah Hotham, Daniel Michelson.

**Writing – review & editing:** Clio Berry, Julia Fountain, Lindsay Forbes, Leanne Bogen-Johnston, Abigail Thomson, Yelena Zylko, Alice Tunks, Sarah Hotham, Daniel Michelson.

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
