## [Decision Letter · Decision Letter 0]

16 Nov 2023

PONE-D-23-17074Developing a hope-focused intervention to prevent mental health problems and improve social outcomes for young women who are not in education, employment or training (NEET): A co-design study in deprived coastal communities in South-East EnglandPLOS ONE

Dear Dr. Berry,

Thank you for submitting your manuscript to PLOS ONE. After careful consideration, we feel that it has merit but does not fully meet PLOS ONE’s publication criteria as it currently stands. Therefore, we invite you to submit a revised version of the manuscript that addresses the points raised during the review process.

We look forward to receiving your revised manuscript.

Kind regards,

Ali B. Mahmoud, Ph.D.

Academic Editor

PLOS ONE

Journal Requirements:

Reviewers' comments:

Reviewer's Responses to Questions

**Comments to the Author**

1. Is the manuscript technically sound, and do the data support the conclusions?

Reviewer #1: Yes

2. Has the statistical analysis been performed appropriately and rigorously? 

Reviewer #1: N/A

3. Have the authors made all data underlying the findings in their manuscript fully available?

Reviewer #1: No

4. Is the manuscript presented in an intelligible fashion and written in standard English?

Reviewer #1: Yes

5. Review Comments to the Author

Reviewer #1: Thank you for giving me the opportunity to review this article titled ‘Developing a hope-focused intervention to prevent mental health problems and improve social outcomes for young women who are not in education, employment or training (NEET): A co-design study in deprived coastal communities in South-East England. The manuscript was interesting to read, and I commend the authors for their participatory attempt to co-design an intervention for NEET young women. As stated, the co-design approach and the explicit focus on NEET young women contributes to the novelty of the research. However, in its current form I think the article needs more work and I base my opinion on the following issues.

Overarching/broader comments

1. The article is well-written and interesting, detailing the iterative developmental approach of identify local population needs and intervention requirements; select and contextualise promising hope-focused practice components; and articulate how a hope-focused intervention might be implemented and scaled up. However, it is currently written in a very linear and rigid IMRAD format, which I do not think makes the results or the process justice. It becomes somewhat descriptive and fragmented, with the key findings being presented (hidden) in tables. Overall, my suggestion would be to write an integrated results and discussion section that narratively describes and elaborates on information in tables 2 (and 3).

a. I would like to have seen the main outcome – the ToC – take center stage and then the authors could “move backwards” to explain how it came to be developed building on findings from phases 1 and 2.

b. This also means that participant characteristics could preferably be presented and summarised in the participant section while the research questions are removed in favor of a clearly articulated (and novel) aim that relates to the co-design of a hope-focused intervention for NEET young women.

2. The data analysis process should be clarified, especially when it comes to the use of critical realism. For me it is not evident how a “critical realist epistemology” aligns with a framework charting approach and the TIDieR framework, and whether the authors did, in fact, elucidate any demi-regularities.

3. I know that the study is written for an academic audience and aimed at detailing the design process, but the practice and action-oriented focus of the study is somewhat lost since a forthcoming implementation is not mentioned. I would very much like to see at least some reflections about the next steps for HOPEFUL, beyond the statement that future evaluation research is needed (which requires some form of implementation to be possible).

Additional comments

4. There is a mixture in reference style, combining Vancover with Harvard (e.g., on page 3).

6. PLOS authors have the option to publish the peer review history of their article (what does this mean?). If published, this will include your full peer review and any attached files.

Reviewer #1: No

---

## [Author Response · Author response to Decision Letter 0]

1 Feb 2024

Review Comments to the Author

Reviewer #1: Thank you for giving me the opportunity to review this article titled ‘Developing a hope-focused intervention to prevent mental health problems and improve social outcomes for young women who are not in education, employment or training (NEET): A co-design study in deprived coastal communities in South-East England. The manuscript was interesting to read, and I commend the authors for their participatory attempt to co-design an intervention for NEET young women. As stated, the co-design approach and the explicit focus on NEET young women contributes to the novelty of the research. However, in its current form I think the article needs more work and I base my opinion on the following issues.

Overarching/broader comments

1. The article is well-written and interesting, detailing the iterative developmental approach of identify local population needs and intervention requirements; select and contextualise promising hope-focused practice components; and articulate how a hope-focused intervention might be implemented and scaled up. However, it is currently written in a very linear and rigid IMRAD format, which I do not think makes the results or the process justice. It becomes somewhat descriptive and fragmented, with the key findings being presented (hidden) in tables. Overall, my suggestion would be to write an integrated results and discussion section that narratively describes and elaborates on information in tables 2 (and 3). 

a. I would like to have seen the main outcome – the ToC – take center stage and then the authors could “move backwards” to explain how it came to be developed building on findings from phases 1 and 2.

b. This also means that participant characteristics could preferably be presented and summarised in the participant section while the research questions are removed in favor of a clearly articulated (and novel) aim that relates to the co-design of a hope-focused intervention for NEET young women.

Response: We thank the reviewer for the very positive comments about our manuscript. We understand and agree with the need to reduce the sense of fragmentation noted with respect to the presentation of our results. In response, we have moved participant details as suggested from results to the Materials and methods section. We have streamlined the section on the aims of the current research - although we have retained our research questions as these were integral to how we approached the study and the analysis, as detailed. We have combined the findings from phases 1 and 2 in the Results section. We understand the reviewer’s point regarding the rigidity of the IMRAD format. We have retained the presentation of the intervention blueprint and theory of change (we position both as the main outcomes/products of the work) as the culmination of the Results section. To present these products earlier we think risks undermining the clarity with which readers understand the iterative developmental approach we took, which the reviewer also agrees is the strength of this work. We also think these outputs follow logically from the data presented. In keeping with our person-based and partnership ethos, we want to ensure transparency with respect to how we combined the data and our interpretations and moved from these to the products of our research. 

2. The data analysis process should be clarified, especially when it comes to the use of critical realism. For me it is not evident how a “critical realist epistemology” aligns with a framework charting approach and the TIDieR framework, and whether the authors did, in fact, elucidate any demi-regularities.

Response: We thank the reviewer for identifying this area for development. We have provided further rationale in the Materials and methods section for the analytic approach taken, especially the alignment between a critical realist stance and framework analysis approach. We have additionally clarified in the Materials and methods section that the integration of findings (i.e., Table 3) across study phases, and with prior evidence, was done with the purpose of identifying demi-regularities (i.e., points of convergence across evidentiary sources) and points of divergence.

3. I know that the study is written for an academic audience and aimed at detailing the design process, but the practice and action-oriented focus of the study is somewhat lost since a forthcoming implementation is not mentioned. I would very much like to see at least some reflections about the next steps for HOPEFUL, beyond the statement that future evaluation research is needed (which requires some form of implementation to be possible).

Response: We thank the reviewer for identifying this as an important addition. The next step for HOPEFUL is evaluation research. We have provided further detail regarding the design of the intervention with scalable implementation in mind, notable points for implementation, and also regarding practical considerations relevant to this. We have additionally emphasised the relevant wider learning from this study for existing provision for young women. 

Additional comments

4. There is a mixture in reference style, combining Vancover with Harvard (e.g., on page 3).

Response: This has been corrected.

---

## [Decision Letter · Decision Letter 1]

14 May 2024

Developing a hope-focused intervention to prevent mental health problems and improve social outcomes for young women who are not in education, employment, or training (NEET): A co-design study in deprived coastal communities in South-East England.

PONE-D-23-17074R1

Dear Dr. Berry,

We’re pleased to inform you that your manuscript has been judged scientifically suitable for publication and will be formally accepted for publication once it meets all outstanding technical requirements.

Kind regards,

Ali B. Mahmoud, Ph.D.

Academic Editor

PLOS ONE

Additional Editor Comments (optional):

Reviewers' comments:

Reviewer's Responses to Questions

**Comments to the Author**

1. If the authors have adequately addressed your comments raised in a previous round of review and you feel that this manuscript is now acceptable for publication, you may indicate that here to bypass the “Comments to the Author” section, enter your conflict of interest statement in the “Confidential to Editor” section, and submit your "Accept" recommendation.

Reviewer #1: All comments have been addressed

2. Is the manuscript technically sound, and do the data support the conclusions?

Reviewer #1: Yes

3. Has the statistical analysis been performed appropriately and rigorously? 

Reviewer #1: N/A

4. Have the authors made all data underlying the findings in their manuscript fully available?

Reviewer #1: No

5. Is the manuscript presented in an intelligible fashion and written in standard English?

Reviewer #1: Yes

6. Review Comments to the Author

Reviewer #1: Thank you for giving me the opportunity to review this revised version of the article. The authors have properly addressed my comments and I have nothing further to add besides making them aware of the dublicate in references 69 and 70.

7. PLOS authors have the option to publish the peer review history of their article (what does this mean?). If published, this will include your full peer review and any attached files.

Reviewer #1: No

---

## [Editor Report · Acceptance letter]

20 May 2024

PONE-D-23-17074R1 

PLOS ONE

Dear Dr. Berry, 

I'm pleased to inform you that your manuscript has been deemed suitable for publication in PLOS ONE. Congratulations! Your manuscript is now being handed over to our production team.

Kind regards, 

on behalf of

Dr. Ali B. Mahmoud 

Academic Editor

PLOS ONE